# Nitrogen, Iron, and Zinc Acquisition: Key Nutrients to *Aspergillus fumigatus* Virulence

**DOI:** 10.3390/jof7070518

**Published:** 2021-06-28

**Authors:** Uxue Perez-Cuesta, Xabier Guruceaga, Saioa Cendon-Sanchez, Eduardo Pelegri-Martinez, Fernando L. Hernando, Andoni Ramirez-Garcia, Ana Abad-Diaz-de-Cerio, Aitor Rementeria

**Affiliations:** Fungal and Bacterial Biomics Research Group, Department of Immunology, Microbiology, and Parasitology, University of the Basque Country (UPV/EHU), 48940 Leioa, Spain; uxue.perezc@ehu.eus (U.P.-C.); xabier.guruceaga@ehu.eus (X.G.); saioa.cendon@ehu.eus (S.C.-S.); epelegri@alumni.unav.es (E.P.-M.); fl.hernando@ehu.eus (F.L.H.); ana.abad@ehu.eus (A.A.-D.-d.-C.)

**Keywords:** *Aspergillus fumigatus*, nitrogen, iron, zinc, nutrient acquisition, siderophores, metals, transcription factors, primary metabolism, nitrogen metabolite repression

## Abstract

*Aspergillus fumigatus* is a ubiquitous soil decomposer and an opportunistic pathogen that is characterized by its large metabolic machinery for acquiring nutrients from media. Lately, an ever-increasing number of genes involved in fungal nutrition has been associated with its virulence. Of these, nitrogen, iron, and zinc metabolism-related genes are particularly noteworthy, since 78% of them have a direct implication in virulence. In this review, we describe the sensing, uptake and regulation process of the acquisition of these nutrients, the connections between pathways and the virulence-implicated genes. Nevertheless, only 40% of the genes mentioned in this review have been assayed for roles in virulence, leaving a wide field of knowledge that remains uncertain and might offer new therapeutic and diagnostic targets.

## 1. Introduction

*Aspergillus fumigatus* is a saprophytic fungus implicated in nitrogen and carbon recycling. Typically of soil decomposers, it is capable of surviving in a wide range of conditions [1]. It has evolved to be a great competitor against other environmental microorganisms because it possesses a controlled nutritional flexibility to scavenge nutrients from the environment efficiently [2]. In addition to these characteristics, its abundant sporulation has gained it a place as one of the most ubiquitous filamentous fungi in the world [3].

Moreover, humans inhale hundreds of tiny hydrophobic conidia from this fungus, which reach the lungs and may cause anything from colonization to invasive aspergillosis, a disease with a mortality rate of up to 95% [4,5,6]. The virulence of *A. fumigatus* depends on both the immune state of the host and the pathogenic capacity of the fungus, the latter being multifactorial and polygenic [7]. Virulence factors previously described in the literature include genes involved in cell wall composition maintenance, such as melanin biosynthetic genes [8] or Galactosaminogalactan [9]; genes associated with combatting the immune system, such as superoxide dismutase [10]; and biosynthetic clusters of genes that encode different toxins, such as gliotoxin [11] or fumagillin [12,13], which are part of a complex secondary metabolism of this fungus.

In addition, the capacity to acquire nutrients such as nitrogen, iron, and zinc is beginning to be recognized as an important element in the pathobiology of the fungus and is even considered a virulence factor [14]. Indeed, the mechanisms of nutrient uptake are of paramount importance in developing an infection since the host is able actively to reduce the availability of nutrients for pathogens [15]. This is the case, for example, with the protein ovotransferrin, which prevents microbial growth in eggs through the chelation of iron [15]. This strategy of regulating nutrients, known as nutritional immunity, is recognized as a third type of immunity, together with humoral and cellular immunity, and involves immune and non-immune cells, molecules, and intra- and extracellular mechanisms [15,16,17]. In consequence, an underlying disease that affects this type of immunity could cause a predisposition to *A. fumigatus* infection [14].

In this review, we seek to compile and integrate the metabolic role of nitrogen, iron, and zinc in the growth of *A. fumigatus* and their involvement in fungal virulence. To this end, we summarize the processes of sensing, uptake, and regulation of acquisition of those nutrients and focus on the genes and proteins that affect pathogenicity, which could be potential therapeutic targets.

## 2. Nitrogen Metabolism

Nitrogen is an essential compound for forming biological structures such as proteins and nucleic acids. It is also one of the main nutrients needed by *A. fumigatus* to survive [18]. However, it is one of the environmental constraints, since not all organisms are capable of assimilating the inorganic forms: atmospheric nitrogen (N_2_), ammonia (NH_4_), nitrite (NO_2_^−^), and nitrate (NO_3_^−^). Fungi, for example, cannot fix N_2_, but they have a very versatile nitrogen metabolism to absorb other organic and inorganic forms of nitrogen, including toxic forms.

There are three possible situations related to this nutrient that *A. fumigatus* may encounter in the environment: (1) niches with easily assimilable forms of nitrogen, (2) niches rich in complex nitrogen compounds, and (3) niches without accessible forms of nitrogen. To deal with all these environmental situations, *A. fumigatus* has diverse metabolic machinery [18]. Activation of the genes involved in the pathway required for nitrogen anabolism or catabolism is dependent on the nitrogen source. Thus, sensing the available compound is the essential signal for activating the specific pathways and, in this way, preventing energy waste. The activation process, which involves triggering gene expression and metabolic cascades [19], is known as nitrogen metabolite repression (NMR) [20], and Table 1 shows the genes involved in this nitrogen metabolism.

### 2.1. Primary Nitrogen Sources: Ammonium and Glutamine Assimilation

The most easily assimilable forms of nitrogen are ammonium and glutamine since their uptake involves low energy costs [20]. In yeasts such as *Candida albicans, Schizosaccharomyces pombe,* and *Cryptococcus neoformans,* ammonium permeases (Mep1 and Mep2 in *C. albicans*; Amt1 and Amt2 in *S. pombe* and *C. neoformans*) allow fungi to grow at limiting concentrations of ammonia [34]. In this regard, the *A. fumigatus* genome has two homologous genes, *mepA* (Afu1g10930) and *meaA* (Afu2g058800) (Table 1), but their role has yet to be studied. Once inside the cytosol, the glutamine synthetase catalyzes the condensation of ammonia and glutamate to produce glutamine, and glutamate synthetase transforms it with 2-oxoglutarate into two molecules of glutamate. Moreover, glutamate dehydrogenase (GdhA) can add ammonia to 2-oxoglutarate to obtain glutamate, acting as the link between nitrogen and carbon metabolism [35] (Figure 1).

Regarding the other primary nitrogen source, glutamine, several different amino acid sensors/permeases (AAPs), such as Csy1p [36] and Gap2p [37], participate in glutamine assimilation and have been related to pseudohyphal growth in yeasts. However, despite the abundance of AAPs described in the *A. fumigatus* genome, including the Afu7g04290 gene (an orthologue of Gap2p), none of them has been characterized [34].

### 2.2. Nitrate and Other Secondary Nitrogen Sources

Nitrate, amides, purines, amino acids, and complex substrates such as collagen and elastin are considered to be secondary nitrogen sources [19]. Of these, nitrate deserves particular mention since its assimilation is part of the global nitrogen cycle and the pathways involved in this process are of great ecological interest. Specifically, the nitrate-assimilatory gene cluster that encodes the three enzymes of the pathway (the nitrate-specific transporter CrnA, the nitrate reductase NiaD, and the nitrite reductase NiiA) have been studied in *A. nidulans* and genetically characterized in *A. fumigatus* [21,22]. The metabolic pathway begins with the entry of nitrate into the cell through CrnA. Once inside the cell, NiaD reduces nitrate to nitrite, and subsequently, NiiA reduces nitrite to ammonium. Finally, ammonium is included in the structure of the glutamine or glutamate, as explained above (Figure 1).

With regard to other secondary nitrogen sources, environmental amides, such as formamide or acetamide, are hydrolyzed by FmdS and AmdS, respectively, releasing ammonium [38,39] which can be integrated into the metabolism as described above. However, these genes have only been investigated in *A. nidulans*. Moreover, *A. fumigatus*, as with other *Ascomycota*, can use purines but not pyrimidines as a nitrogen source. For this purpose, the *A. fumigatus* genome encodes three known purine/H+ symporters: UapC, AzgA, and FcyB [23]. Thus, purines enter the purine cycle directly and come out as urea, which is easily transformed into ammonium by ureases [18] (Figure 1).

On the other hand, when complex proteins are the nitrogen source, *A. fumigatus* is able to promote protease secretion specific to the protein substrate [40]. After digestion, the fungus could assimilate nitrogen in the form of amino acids or peptides through AAPs. These amino acids enter the Krebs cycle or directly form new proteins.

### 2.3. Nitrogen Metabolite Repression (NMR) as a Regulatory Mechanism

Due to the numerous pathways and transcription factors (TFs) involved in NMR, the study of this regulatory mechanism presents a difficult challenge. The principal function of NMR is to prioritize the use of easily assimilated nitrogen forms and the main regulatory gene GATA-like TF, *areA*, regulates nitrogen catabolism following this principle [24] (Figure 1). This TF is involved in all the aforementioned assimilation pathways since, in the Δ*areA* deletion mutant, the expression of many nitrogen catabolic genes are affected—namely, *amdS* [39] and *fmdS* [38] for amides; *gdhA* [41] for ammonium; *hxA, uaZ*, and *uapA* [42,43] for purines and *crnA* and *niaD* [21] for nitrate. Consequently, this Δ*areA* mutant can only use ammonium or glutamine as a nitrogen source [44].

In addition, regulators of the *areA* gene have been found in *A. nidulans*, and they show a high degree of homology in *A. fumigatus*. For example, when the media has primary nitrogen sources, the co-repressor NmrA interacts with AreA to inhibit the induction of genes involved in the assimilation of secondary nitrogen sources [45]. Furthermore, the presence of intracellular glutamine represses the areA expression via degradation of *areA* mRNA [35]. However, in the absence of primary nitrogen sources, TamA is the co-activator that interacts with AreA, thus enhancing the activation of nitrogen catabolism [41,46]. Indeed, with secondary nitrogen sources, each pathway is activated by the substrate and the interaction between AreA and the substrate pathway-specific TF activate only the pathway required for the available nitrogen source [18,44,47]. For example, the presence of nitrate activates its assimilatory pathway through the interaction between the TF NirA and AreA [19].

The mechanism for sensing nitrogen needs to be studied in depth since it is the first signal to activate nitrogen metabolism, but most sensors, such as ammonium permeases and AAPs, have not been investigated in *A. fumigatus*. On the other hand, the target of the rapamycin (TOR) pathway is a conserved nitrogen sensor, and the implication of the *tor* gene (Afu2g10270) in nitrogen metabolism has been demonstrated in *A. fumigatus* [25]. In *S. cerevisiae,* the corresponding *tor* and *areA* homolog genes interact in response to nutrient availability [48]. The Ras-like small GTPase RhbA appears to be involved in this pathway and collaborates in response to changes in nitrogen concentration, especially during starvation [26]. Other conserved pathways that are also activated by environmental changes are the mitogen-activated protein kinase (MAPK) cascades. A good example is MAP kinase SakA (codified by the Afu1g12940 gene), which is involved in the nitrate and nitrite assimilation cascade during the germination process, since the Δ*sakA* deletion mutant germinates later than the wild type (WT) strain in the presence of both nitrogen sources [27]. There are other conserved signaling pathways, such as cAMP, Wnt, Ras, and others, that might be involved in nitrogen sensing. However, although this is part of the primary metabolism, little is known about the process, and it needs to be studied in greater depth.

The NMR is responsible for regulating the secretion of proteases to digest and assimilate complex extracellular nitrogen sources. Thus, protease secretion is mediated via the TF PrtT, achieving secretion of a pattern of proteases specific to the nitrogen source present in the media [28].

In contrast to the positive regulatory mechanisms described in previous paragraphs, negative regulation of nitrogen metabolisms is performed by AreB, the antagonist of AreA, which appears to act in nitrogen or carbon-limiting conditions. However, its mode of action has to be elucidated [49,50].

In addition to the regulation mechanism included in NMR, it is worth mentioning that the TF CpcA regulates amino acid biosynthesis through cross-pathway control (CPC), but little is known about its action mechanisms [30]. For example, amino acid starvation triggers the phosphorylation of the α subunit of eIF2 by the sensor kinase CpcC and results in the translation of *cpcA* to generate a cellular response to starvation [29] (Figure 1).

### 2.4. Importance of Nitrogen Acquisition for A. fumigatus Virulence

In the human body, *A. fumigatus* usually needs to secrete proteases to obtain the nitrogen of proteins and cofactors, but free nitrogen forms such as ammonium and glutamine are also present in blood and tissues [34]. Although it seems clear that the NMR is necessary during infection, as demonstrated by the fact that the establishment of pulmonary aspergillosis was delayed in mice infected with the Δ*areA* strain [24], little is known of the importance of the genes involved in the assimilation and metabolization of these nitrogen sources in virulence. Only 23% of them have been described as being required for *A. fumigatus* virulence, such as *rhbA* [26] (Figure 2).

Another example of the importance of nitrogen metabolism regulation in virulence comes from the CPC system, which appears to be essential for virulence since the *A. fumigatus* Δ*cpcA* deletion strain was avirulent in a murine model of pulmonary aspergillosis [30]. In contrast, the Δ*cpcC* strain retained its normal virulence, which could mean that other genes may activate CpcA [29,30]. Concerning the regulation of nitrogen metabolism, a TOR repression-like pattern is observed during initiation of neutropenic CD1 male mice infection [51], i.e., a strong increase of the catabolism of carbohydrates and nitrogen.

During infection, *A. fumigatus* secretes proteases that destroy the lung epithelium. The release of amino acids can be used as a source of carbon and nitrogen. In this way, amino acid permeases are upregulated in contrast to nitrate assimilation in a murine infection model [51]. However, early in the infection process, the catabolism of amino acids is upregulated to produce energy, as mentioned above. On the other hand, some amino acid biosynthetic pathways, such as histidine and/or lysine, may be implicated in *A. fumigatus* virulence. Indeed, the *A. fumigatus* Δ*hisB* auxotroph strain showed attenuated pathogenicity in four infection models [31]. Moreover, the Δ*lysF* mutant strain, which lacks the enzyme of the last step of lysine biosynthesis, is also less virulent in a murine model [32], in contrast to what occurs in *A. nidulans* [19]. Both amino acids are present in the lungs but may be below the requirements of *A. fumigatus*. Furthermore, studies have described the close relationship between L-amino acids metabolism and the *Aspergillus* genus germination process [52,53].

Apart from these examples, the role of virulence in most of the genes involved in nitrogen metabolism has yet to be studied, or they are not implicated in virulence. As can be seen in a more illustrative form in Figure 2, only 36% of nitrogen metabolism-implicated genes have been assayed in virulence, and 44% of them are already putative genes.

Given all of the above, we may conclude that nitrogen is an essential nutrient for *A. fumigatus*, and this compound and its metabolism are involved in the synthesis of a large proportion of cellular components, with a very complex regulation process. However, little is known about its functioning, and more in-depth study would be of great assistance. Indeed, identifying the differences between *A. nidulans* and *A. fumigatus* might provide the keys to understanding *A. fumigatus* pathogenicity. After all, successful metabolic adaptation to the host is crucial to fungal growth during human infections.

## 3. Iron Metabolism

Iron is the most abundant trace metal in organisms and an indispensable cofactor for many essential metabolic processes, including the electron transport chain, amino acid metabolism, DNA biosynthesis, sterol formation, and oxidative stress detoxification [17,54,55]. The presence and availability of this metal in the environment are dependent on the type of atmosphere and the pH. Indeed, the pH is the determinant for iron solubility, since it is soluble in aerobiosis in acid conditions, but with neutral to alkaline pH, it tends to precipitate. Moreover, although the ferrous form (Fe^2+^) can be more readily captured, due to its higher solubility, the ferric ion (Fe^3+^) is more abundant.

*A. fumigatus*, such as other fungi and bacteria, can obtain iron and maintain its homeostasis inside the fungal cell in different ways [17,56,57]. Further, the balance between the acquisition, storage, and consumption of iron is crucial for fungal survival [54] since, although iron is essential, excess results in the generation of ROS through a Fenton/Haber–Weiss reaction and the inactivation of certain enzymes by replacing other metal cofactors [17]. Table 2 shows the genes involved in iron metabolism in *A. fumigatus*.

### 3.1. Mechanisms for Iron Acquisition and Storage in A. fumigatus

The different forms in which iron occurs in the environment require different strategies of acquisition. Specifically, *A. fumigatus* has two high-affinity mechanisms, reductive iron assimilation (RIA) and siderophore-mediated uptake [58], and one low-affinity system conducted by general metal permeases [69].

Of these systems, RIA is a highly conserved strategy in fungal species for Fe^3+^ intake [17]. This acquisition process starts with membrane metalloreductases that reduce Fe^3+^ to soluble form Fe^2+^ and the ferroxidase FetC then re-oxidizes it to allow transport by the iron permease FtrA [58] (Figure 3). It appears that this mechanism may be redundant, as 15 putative membrane metalloreductases have been described in the *A. fumigatus* genome.

On the other hand, siderophores are low-mass organic Fe^3+^ chelators [54,56,58], which have six oxygen atoms in an octahedral geometry to avoid ion reduction and prevent ROS formation [64]. Specifically, *A. fumigatus* produces four hydroxamate type siderophores, two extracellular—fusarinine C (FsC) and its derivative triacetylfusarinine C (TAFC)— and two intracellular—ferricrocin (FC) and hydroxyferricrocin (HFC) [59]. They are all derived from L-ornithine, mainly from the mitochondria [70], which forms N5-hydroxy-L-ornithine through the hydroxylation enzyme SidA. Later, an acyl group is added to form the iron-binding hydroxamate group [71]. Depending on the acyl residue, either fusarinine-type or ferricrocin-type siderophore is formed [59,62] (Figure 3). Each siderophore contains three hydroxamate groups linked by ester or peptide bonds to form a hexadentate structure, increasing Fe^3+^ affinity [72].

Extracellular siderophores take the extracellular Fe^3+^ and cross the fungal membrane via siderophore-iron transporters (SITs). This uptake mechanism is highly conserved in fungi, and even non-siderophore-producing species, such as *S. cerevisiae, Candida spp.* [73], and *C. neoformans* [74], have SITs to use xenosiderophores (siderophore used by an organism other than the one that produces it). Likewise, in *A. fumigatus*, there are seven putative SITs to transport their own siderophores and some xenosiderophores [75]. Once inside the cytosol, hydrolysis of the siderophores releases the iron for utilization or storage. The siderophore-degrading enzyme EstB (a bacterial homolog enzyme [63]) and the SidJ [61] are involved in the degradation of TAFC and FsC, respectively (Figure 3). However, they are not the only degrading enzymes involved in these processes since deleting them only partially reduces siderophore degradation [64].

Ferritin is the protein responsible for iron storage in bacteria, plants, and animals. In contrast, fungi follow a different strategy based on iron accumulation in the vacuole and intracellular siderophores due to the absence of ferritin in this kingdom [54]. In *A. fumigatus*, the iron enters the vacuole via the iron importer CccA, homologous to *S. cerevisiae* Ccc1 [69]. *S. cerevisiae* employs the Ccc1 not only to import iron to vacuoles but also to expel the iron from the vacuole and recycle it. However, in *A. fumigatus,* there must be an alternative, unknown mechanism or an uncharacterized transporter since CccA is unable to recycle this metal [64] (Figure 3). In the case of intracellular siderophore, FC and HFC are used for hyphal or conidial iron storage, respectively [59]. In addition, under iron-depleted conditions, FC is implicated in the normal growth of hyphae and sporulation, and HFC is essential to break up the dormancy and swell and form the germ tube [59].

### 3.2. Regulatory Mechanisms for Iron Homeostasis

Since no iron-excretion mechanism is known in *A. fumigatus*, control of iron uptake is the main regulatory mechanism for maintaining iron homeostasis [75]. Two main repressors for this purpose have been described: the GATA TF SreA, which downregulates acquisition [65], and the HapX to downregulate iron consumption [66]. During iron sufficiency, SreA represses high-affinity iron uptake and upregulates the consumption of iron via heme biosynthesis, respiration, and ribosome biogenesis [65,76], and HapX upregulates vacuolar storage [77]. During iron starvation, HapX interacts with the DNA-binding CCAAT-binding complex (CBC) to activate RIA, upregulates the synthesis and uptake of siderophores, and represses the iron-consuming pathways [66,77,78]. At the same time, HapX and SreA have negative feedback regulation due to their antagonistic functions [66] (Figure 3). In addition, both appear to be post-translationally regulated in opposing ways by the presence of iron, blocking HapX and activating the SreA function [78].

Apart from the specific regulation, iron metabolism is coordinated with other metabolic pathways. As iron is necessary for glycolysis, the TCA cycle, and the electron transport chain, cellular iron increases when these metabolisms are upregulated [79]. One such situation is during hypoxia, when SrbA, a TF essential for hypoxic growth, activates siderophore iron uptake, partly by transcriptional activation of the *hapX* gene [80] (Figure 3). Gluconeogenesis is also linked to iron metabolism since the TFs AcuM and AcuK, both involved in this pathway, repress the *sreA* gene, inducing iron acquisition via high-affinity iron uptake [67,68] (Figure 3).

Another factor involved in iron regulation is pH, since (as explained above) alkaline conditions decrease iron solubility. In *A. nidulans* TF PacC upregulates siderophore biosynthesis in neutral pH as opposed to acidic conditions [81], and the *A. fumigatus* homolog protein PacC might play a similar role.

### 3.3. Importance of Iron Metabolism for A. fumigatus Virulence

Regarding fungal pathogenesis, iron is one of the keys to *A. fumigatus* virulence. In the human body, iron is sequestered by hemoglobin, hepcidin, lactoferrin, transferrin, etc. [82], and during infection, it is actively restricted even further as a defense mechanism of the innate immune system [83]. Moreover, although *A. fumigatus* is well-adapted to cause human infection, it is not capable of uptaking iron from sources such as ferritin or transferrin [58] or in the case of hemoglobin only when its concentration is very high [84], so the fungus has to adapt to starvation during infection. In consequence, people with increased free iron in tissues—as a result of a range of diseases, such as hematopoietic malignancies or even liver transplantation—have a higher risk of suffering invasive aspergillosis [84,85,86,87].

With regard to iron uptake, the siderophore-mediated system appears to be more relevant than RIA during infections because the Δ*sidA* mutant strain, which is unable to produce siderophores, had attenuated virulence, while the Δ*ftrA* mutant strain, which lacks the iron permease of RIA, had similar virulence to WT in the murine model of invasive aspergillosis [58]. Nonetheless, RIA components are induced during the initiation of infection to support siderophore-mediated acquisition [51]. The Δ*sidA/*Δ*ftrA* double mutant is unable to grow unless the medium is supplemented with FC or high levels of Fe^2+^ [58]. In addition, other mutant strains incapable of producing extracellular siderophores, such as the Δ*sidD,* Δ*sidF,* Δ*sidH,* and Δ*sidI* strains, and intracellular siderophore, such as the Δ*sidC* strain, have been shown to have attenuated virulence in the immunosuppressed murine model, and both types of siderophores appear to be virulence factors [59,60]. As an exception, only a TAFC non-producer Δ*sidG* strain was as virulent as the WT [59] (Figure 2).

In a different infection model of fungal keratitis, only the absence of extracellular siderophores inhibits fungal growth [88]. Indeed, the addition of deferoxamine, an iron chelator which *A. fumigatus* can use as a xenoxiderophore, causes an increase in the fungal mass, while the addition of deferiprone, an iron chelator that cannot be used by *A. fumigatus*, inhibits fungal growth [88].

Siderophore biosynthesis is also related to the growth of conidia inside host cells. Indeed, defects in its biosynthesis lead to a reduction in intracellular growth and survival of *A. fumigatus* after phagocytosis by murine alveolar macrophages [89]. Moreover, the immune response induced by the contact with WT or with siderophore non-producer strains in these cells was different [90].

The TFs related to iron homeostasis also play a role in the virulence of *A. fumigatus*, highlighting the importance of this process in virulence. Indeed, inhibition of both HapX and SreA is lethal for the fungus [65,66,78], and the absence of HapX but not SreA attenuates virulence in the murine model of aspergillosis [65,66] underlining the critical role of adaptation to iron limitation in virulence.

In addition to nutritional immunity, polymicrobial competition for iron may also limit availability. In cystic fibrosis patients, bacteria are predominant, and it has been demonstrated that *Pseudomonas aeruginosa* uses iron competition to inhibit *A. fumigatus* growth. In particular, this bacteria produces pyoverdine and pyochelin, iron chelators, and the bacteriophage Pf4, which inhibits fungal metabolism, but its effect could be overcome with ferric ion supplementation [91].

In conclusion, iron metabolism is one of the most studied metabolic pathways, and it is closely involved in *A. fumigatus* pathogenicity because it is well adapted to the host environment. Unbalancing iron homeostasis could be a good strategy for avoiding fungal infection, and further study of its metabolism may therefore offer new therapeutic strategies.

## 4. Zinc Metabolism

Zinc is the second most prevalent transition metal in cells, after iron, and the second most abundant metal-cofactor of enzymes after magnesium [55]. It therefore plays a crucial role in living organisms. The 40% of proteins that bind to zinc ions are TFs, and many metabolic pathways are affected by zinc depletion [55,92]. Furthermore, when the zinc concentration is below the minimum quota for supplying enzymes, cell growth and development are not possible [93]. In contrast, an excess of this metal causes toxicity by replacing other metals in protein-metalation sites. Although, zinc does not damage cells through ROS (as iron excess does) because it is a redox-inactive metal [17]. Table 3 shows the genes involved in the metabolism of this metal.

### 4.1. Zinc Acquisition and Homeostasis

This metal is found in the environment in the form of zinc ion (Zn^2+^), and *A. fumigatus* has several advanced mechanisms for uptaking it since zinc availability is dependent on the acidity of the media, such as iron [99]. Specifically, *A. fumigatus* has eight genes that encode Zrt-/Irt-like proteins (ZIP) transporters, involved in zinc uptake, but only three have been studied: ZrfA, ZrfB [94], and ZrfC [95] (Figure 4). Once Zn^2+^ is inside the cytosol, the cell employs this ion in numerous metabolic pathways such as endoplasmic reticulum function, oxidative stress resistance, protein folding and synthesis, vesicular trafficking, and chromatin modification [17,100].

When the zinc concentration exceeds cellular requirements, the vacuole acts as the storage organelle, as in the case of iron. The machinery involved in zinc homeostasis is well known in *S. cerevisiae* and, using homology, more than 30 genes can be predicted to be involved in zinc homeostasis of *A. fumigatus*. Of these, there are eight encode cation diffusion facilitator (CDF) exporters implicated in zinc detoxification: Afu1g12090, Afu6g14170, Afu1g14440, Afu6g00440, Afu7g06570, Afu4g04150, Afu2g14570, and Afu5g09830, although none of them have been studied [101] (Table 3, Figure 2).

### 4.2. Regulatory Mechanisms

The efficient uptake of zinc in *A. fumigatus* is the responsibility of two TFs that sense both zinc concentration and pH. The TF ZafA is responsible for upregulating *zrfA, zrfB*, and *zrfC* genes under zinc-limiting conditions and is inhibited by the presence of Zn^2+^ [96]. The TF PacC, on the other hand, inhibits *zrfC* gene expression in acid conditions [95] and inhibits *zrfA* and *zrfB* genes in neutral to alkaline conditions [97] (Figure 4). Indeed, a Δ*zrfC* deletion mutant did not grow in Zn-limiting alkaline conditions, a Δ*zrfAB* double-deletion mutant was incapable of growing under Zn-limiting acid conditions, and a Δ*zrfABC* triple-deletion mutant was unable to grow in any Zn-limiting conditions [95].

On the other hand, the TF ZafA and the allergen Aspf2 had previously been referenced in the literature. This allergen is expressed in alkaline zinc-limiting conditions under the control of ZafA and could help to supply Zn to transporters by forming a complex with the ion [95].

### 4.3. Importance of Zn Acquisition for A. fumigatus Virulence

In the human body, zinc is strongly attached to zinc-binding proteins, such as p53 and Myt1, both inside the cells and in extracellular fluids and serum. For example, the average concentration of free zinc is 0.08 µM, 150-fold less than the minimal concentration needed for fungal growth [102,103]. In addition, at physiological pH (7.3–7.4), Zn^2+^ availability is lower than at acid pH [99]. Like iron, there are host defense mechanisms to restrict even further the concentration of Zn^2+^ during fungal infection. For example, when neutrophils are recruited to the site of infection, they release calprotectin, a Zn/Mn-chelating protein [16]. Moreover, the bacterial competition interferes with the availability of Zn since pyoverdine and pyochelin siderophores are Zn chelators [104].

The genes of *A. fumigatus* involved in Zn acquisition under these conditions are therefore of paramount importance for fungal virulence. This is the case of the *zrfC* gene, which, as previously mentioned, is the key gene upregulated in neutral to alkaline zinc-limiting conditions, as found in the host. Supporting this idea, the Δ*zrfC* strain has reduced virulence in both the leukopenic and non-leukopenic mice model [103], whereas the Δ*zrfAB* double mutant presents normal virulence [96,103]. Moreover, the Δ*zafA* mutant strain is avirulent since this gene is the regulator of *zrf* gene expression, as well as the Δ*zrfABC* triple mutant strain [96,103] (Figure 2). It is also worth noting that the *zrfC* gene is the only *zrf* gene that has an unusually long N-terminus located on the extracellular side of the membrane to obtain Zn directly from the lungs [103] and that makes *A. fumigatus* resistant to calprotectin since the Δ*zrfC* mutant provokes growth arrest in the presence of this host defense protein [103].

Consequently, the genes of *A. fumigatus* involved in the metabolisms of the essential nutrient Zn might be good targets for therapeutic studies, particularly *zrfC* and *zafA*, which do not have human orthologues and are required to generate a fungal infection.

## 5. Interaction between Metabolic Pathways

Nitrogen, iron, and zinc play an imperative role in *A. fumigatus* adaptation and virulence. This is clearly shown by the diversity of sensing, uptake, and regulatory mechanisms that allow fungal growth in a wide range of conditions, including the host. Understanding the different metabolic pathways, which are interconnected and controlled to work in coordination, helps to interpret the fungal nutritional requirements from a broader point of view. This process begins at the swelling step of germination when the fungus interacts with the environment for the first time [105]. At this point, *A. fumigatus* prioritizes certain metabolic pathways depending on the external signal sensed, involving highly variable molecular responses. One example is the sensing of carbon, the mere presence of which is sufficient to activate the metabolic machinery [106] and, depending on the carbon status of the cell, the TF NmrA could modulate AreA and AreB activity in nitrogen metabolism [50]. It is therefore of paramount importance to integrate the available information and study the metabolic pathways as part of a whole, as this approach is closer to the real infection process.

Nitrogen and iron metabolisms are also connected, since the precursor of siderophores is the amino acid ornithine and, although CPC is the system that regulates all amino acid biosynthetic routes [70], the presence of HapX can increase the ornithine pool inside the cell [66]. Similarly, leucine biosynthesis needs glutamate as a precursor, and the TF LeuB is able to overexpress GdhA to ensure glutamate supply [33], but it also activates siderophore biosynthetic genes and upregulates the *hapX* gene in iron starvation [107]. Consequently, depletion of LeuB causes proteolysis and autophagy, but this effect can be mitigated by supplementing either nitrogen or iron [107]. Indeed, leucine biosynthesis requires an iron–sulfur cluster for two biosynthetic enzymes. This could indicate that iron availability indirectly affects amino acid metabolism. On the other hand, the protease regulator PrtT is able to upregulate iron uptake genes [108].

Concerning iron and zinc metabolisms, they are regulated in a coordinated fashion to avoid toxicity. On the one hand, the intracellular accumulation of zinc decreases the production of extracellular siderophores and, on the other hand, iron starvation downregulates the expression of the importer *zrfB* and upregulates zinc vacuolar storage [109]. Moreover, in iron-limiting conditions, intracellular zinc content is higher than during iron sufficiency, which provokes a zinc hypersensitivity that seems to be an outcome of unbalanced iron homeostasis [109]. The same might occur with other metals such as manganese, whose metabolism is similar to that of zinc [17] and which, as with zinc and iron, is affected by calprotectin [101,110].

An important environmental factor closely related to the pathogenicity of *A. fumigatus* is the pH. The PacC, in addition to regulating pH-dependent zinc and iron uptake, governs the protease secretion required for epithelial damage and, consequently, Δ*pacC* is unable to invade pulmonary epithelium [111]. Furthermore, *A. fumigatus* is capable of alkalinizing the phagolysosomes in a controlled manner, without reaching a pH of over 6.5 (it avoids metal precipitation) to enable efficient iron and zinc uptake [17].

The uptake of nutrients is part of primary metabolism, which is the source of secondary metabolism precursors. One example described is the nitrogen-regulatory *areA* gene, which also modulates aflatoxin production in *Aspergillus flavus* [112]. It would therefore be interesting to determine the interconnections between primary and secondary metabolisms. These might occur at the regulation level, since the global sensing system TOR and MAP-kinases signal pathway are implicated in sensing nitrogen, act upstream of HapX, and thereby regulate siderophore production [25]. Evidence of this interconnection has been established in *Fusarium fusjikuroi,* since TOR controls—partially through AreA—the bikaverin biosynthetic genes [113]. Moreover, *A. fumigatus* produces more gliotoxin in Zn-limiting conditions than in Zn sufficiency and, surprisingly, Δ*zafA* is unable to produce gliotoxin [98]. The regulatory mechanism of gliotoxin biosynthesis by zinc has recently been described and matches what happens during infection. The low concentration of Zn upregulates the *zafA* gene, which, in addition to upregulating Zn-metabolism genes, upregulates expression of the gliotoxin biosynthetic cluster through GliZ activation [98]. The association between a low concentration of Zn and this toxin could be a response to overcome the nutritional and innate immunity of the host, since gliotoxin protects against phagocytosis [114] and Zn depletion is a common mechanism of host defense.

## 6. Conclusions

Despite all the information collated in this review, knowledge of nitrogen, iron, and zinc metabolism is still incomplete. Only 60% of the genes highlighted in this review have been characterized in *A. fumigatus*. Of these confirmed genes, only 40% have been studied for virulence, and most (80%) showed a direct implication in *A. fumigatus* virulence, given that the mutant strains were less virulent, demonstrating that the study of fungal nutrient metabolism is an important field for increasing knowledge of fungal biology.

In the case of nitrogen, only a general idea of how this metabolism works is available. The NMR system needs to be investigated more in depth to understand not only what happens during infection but also to complete the knowledge of the basic metabolism of this fungus. In addition, this may lead to the emergence of new treatment strategies or industrial applications. On the other hand, iron metabolism is well described, and numerous advances have been made over the past few years. Finally, numerous genes related to zinc metabolism have been identified and need to be verified. Moreover, new virulence factors, such as ZrfC, have been described, and clear relationships between zinc and pH have been established.

For future perspectives, comparative studies between different human pathogens may open up new approaches for understanding the metabolism and virulence of *A. fumigatus*. Indeed, the immune barriers faced by bacteria, protozoa, and fungi are similar and, despite having different molecular mechanisms to cope with them, analogous solutions may be found in all groups of pathogens.

To increase knowledge in this area, a comparison with the best-known *Aspergillus* species, *A. nidulans*, may help in extracting a general idea of the molecular mechanisms involved. However, it is also necessary to check these data in *A. fumigatus* because of its greater pathogenicity. This is the case of two examples: the gene lysF, which is essential for *A. fumigatus* virulence but not for *A. nidulans*, and the TF NmrA, which does not have a clear homolog in *A. fumigatus*. Considering this, the differences in basal metabolism between the two species of *Aspergillus* may be crucial for the infection capacity of the fungus.

## Figures and Tables

**Figure 1 jof-07-00518-f001:**
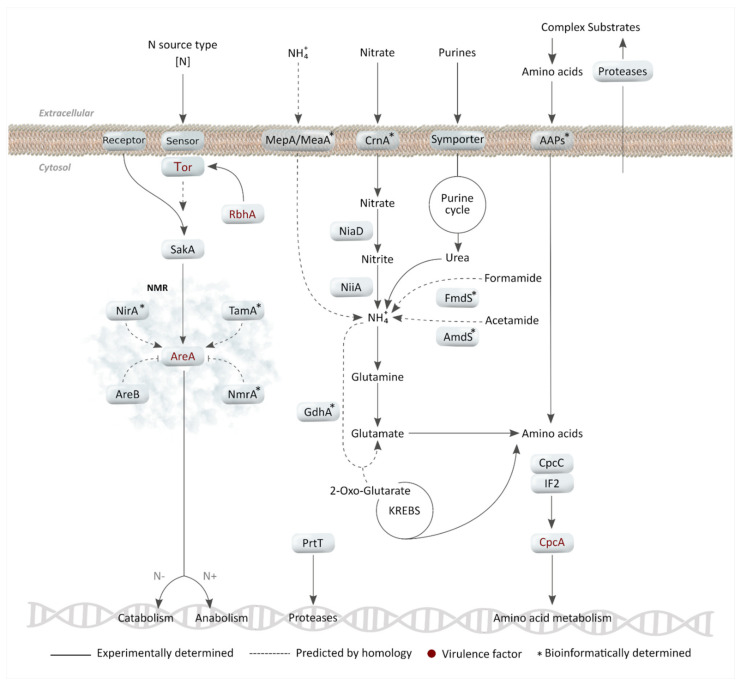
Nitrogen sensing, uptake, metabolism, and regulation in *A. fumigatus*. Each receptor is in charge of sensing and importing a form of extracellular nitrogen. All forms then reach a common step, the transformation to ammonium. This molecule is incorporated into the glutamate pool, the precursor of all other amino acids. Nitrogen metabolite repression (NMR) positively regulates nitrogen catabolism with a lack of nitrogen or presence of second sources of nitrogen and anabolisms when ammonium or glutamine is present in the media. The CPC system regulates amino acid biosynthesis.

**Figure 2 jof-07-00518-f002:**
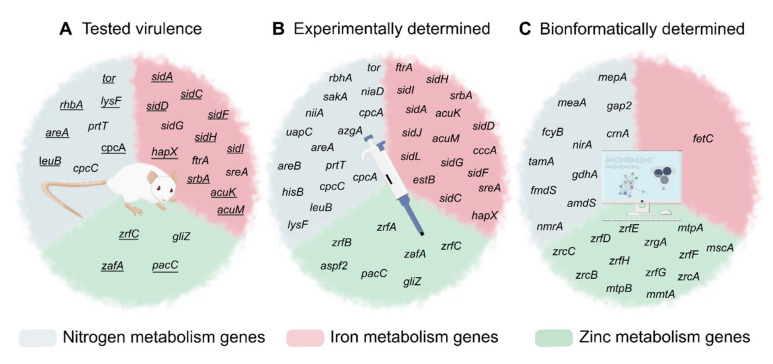
Genes implicated in nitrogen (blue), iron (red), or zinc (green) metabolisms, grouped by current knowledge status. (**A**) Assayed for virulence in cellular, invertebrate, or murine model of infection. The underlined genes present a phenotype with less virulence than WT when they are deleted. (**B**) Genes determined in vitro by expression analysis, enzyme activity assays, etc. (**C**) Genes with predicted function determined by gene or protein homology with other fungal species.

**Figure 3 jof-07-00518-f003:**
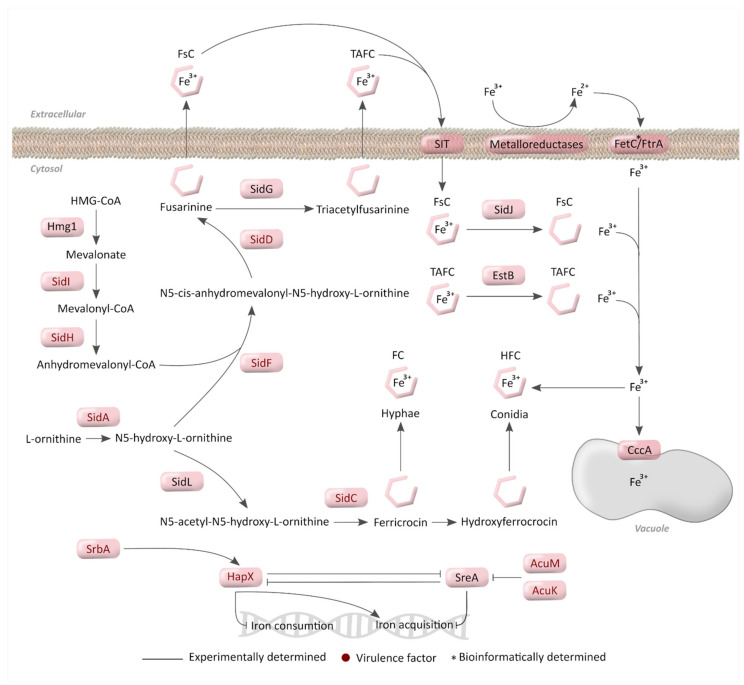
Iron sensing, uptake, metabolism, and regulation in *A. fumigatus*. Iron enters the fungal cell through metal permeases (low affinity), siderophores uptake (high-affinity), or FetC/FtrA protein part of reductive iron assimilation (RIA) (high-affinity). Genes in the Sid family are in charge of the biosynthetic route of extracellular siderophores, FsC and TAFC, and intracellular siderophores, FC and HFC. Siderophores transport Fe^3+^ inside the cell where vacuole or intracellular siderophores store it. HapX and SreA regulate iron to acquire it when necessary and increase its consumption when there is an excess, to avoid metal toxicity.

**Figure 4 jof-07-00518-f004:**
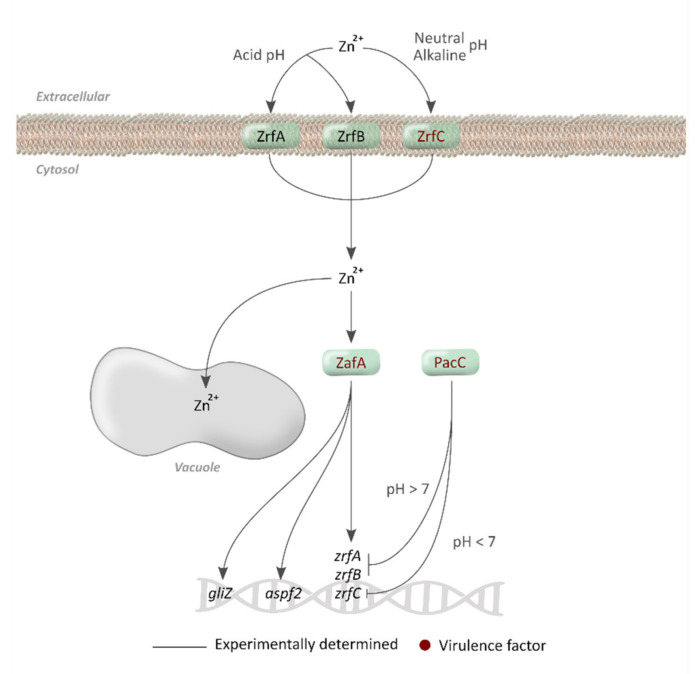
Zinc sensing, uptake, metabolism, and regulation in *A. fumigatus*. Zinc enters the cytosol through ZrfA or ZrfB at acid pH or through ZrfC in neutral to alkaline conditions. It is stored in vacuoles. ZafA TF overexpresses *zrfA, zrfB* and *zrfC* genes in Zn-depleted conditions and PacC inhibits *zrfA/B* in alkaline conditions and *zrfC* in acid pH.

**Table 1 jof-07-00518-t001:** *A. fumigatus* genes implicated in nitrogen metabolism and their implication for fungal phenotype and virulence.

Function	Systematic Name	Standard Name	Phenotype	Virulence	Ref.
Putative ammonium transporter	Afu1g10930	*mepA*	Unknown	Unknown	
Putative ammonium transporter	Afu2g05880	*meaA*	Unknown	Unknown	
Putative amino acid permease	Afu7g04290	*gap2p*	Unknown	Unknown	
Putative nitrate transporter	Afu1g12850	*crnA*	Unknown	Unknown	[21]
Putative nitrate reductase	Afu1g12830	*niaD*	Viable	Unknown	[22]
Putative nitrite reductase	Afu1g12840	*niiA*	Viable	Unknown	[21]
High-affinity purine transporter	Afu7g05910	*uapC*	Viable	Unknown	[23]
High-affinity purine transporter	Afu5g09750	*azgA*	Viable	Unknown	[23]
Putative purine-cytosine permease	Afu2g09860	*fcyB*	Unknown	Unknown	
Putative formamidase	Afu2g02020	*fmdS*	Unknown	Unknown	
General amidase	Afu6g08000	*amdS*	Unknown	Unknown	
Putative glutamate dehydrogenase	Afu4g06620	*gdhA*	Unknown	Unknown	
Nitrogen GATA-like transcription factor	Afu6g01970	*areA*	Decreased utilization of nitrogen sources	Decreased	[24]
Putative nitrogen metabolite repression regulator	Afu5g02920	*nmrA*	Unknown	Unknown	
Putative co-activator of *areA*	Afu3g08050	*tamA*	Unknown	Unknown	
Putative nitrogen GATA-like transcription factor	Afu2g13380	*areB*	Unknown	Unknown	
Putative regulator of nitrate assimilation	Afu5g12020	*nirA*	Unknown	Unknown	
Tor Kinase	Afu2g10270	*tor*	Inviable	Absent	[25]
Ras-like signaling protein	Afu5g05480	*rbhA*	Viable	Decreased	[26]
Mitogen-activated protein kinase	Afu1g12940	*sakA*	Decreased growth and germination	Unknown	[27]
Transcriptional regulator of extracellular proteases	Afu4g10120	*prtT*	Decreased nitrogen utilization and proteases secretion	Normal	[28]
Putative protein kinase for IF2	Afu5g06750	*cpcC*	Decreased resistance to starvation	Normal	[29]
Transcriptional activator of the CPC system	Afu4g12470	*cpcA*	Decreased competitive fitness	Decreased	[30]
Imidazol-glycerol-phosphate dehydratase	Afu6g04700	*hisB*	Histidine auxotrophy	Decreased	[31]
Homoaconitase	Afu5g08890	*lysF*	Lysine auxotrophy	Decreased	[32]
Leucine transcriptional activator	Afu2g03460	*leuB*	Decreased growth without leucine or iron	Decreased	[33]

**Table 2 jof-07-00518-t002:** *A. fumigatus* genes implicated in iron metabolism and their implication for fungal phenotype and virulence.

Function	Systematic Name	Standard Name	Phenotype	Virulence	Ref.
Putative ferroxidase	Afu5g03790	*fetC*	Unknown	Unknown	
High-affinity iron permease	Afu5g03800	*ftrA*	Viable	Normal	[58]
L-ornithine N5-oxygenase	Afu2g07680	*sidA*	Decreased iron utilization and conidiation	Decreased	[58]
Ferricrocin non-ribosomal peptide synthetase	Afu1g17200	*sidC*	Intracellular siderophores non-productive	Decreased	[59]
Fusarinine C non-ribosomal peptide synthetase	Afu3g03420	*sidD*	Extracellular siderophores non-productive	Decreased	[59]
Hydroxyornithine transacylase	Afu3g03400	*sidF*	Extracellular siderophores non-productive	Decreased	[59]
Acetyltransferase	Afu3g03650	*sidG*	Triacylfusarinine non-productive	Normal	[59]
Mevalonyl-CoA hydratase	Afu3g03410	*sidH*	Extracellular siderophores non-productive	Decreased	[60]
Mevalonyl-CoA ligase	Afu1g17190	*sidI*	Extracellular siderophores non-productive	Decreased	[60]
Fusarinine C esterase	Afu3g03390	*sidJ*	Viable	Unknown	[61]
GNAT-type acetyltransferase	Afu1g04450	*sidL*	Intracellular siderophores non-productive	Unknown	[62]
Triacetylfusarinine C esterase	Afu3g03660	*estB*	Viable	Unknown	[63]
Iron vacuolar transporter	Afu4g12530	*cccA*	Decreased iron resistance	Unknown	[64]
Iron GATA transcription factor	Afu5g11260	*sreA*	Decreased iron and oxidative stress resistance	Normal	[65]
Iron bZIP transcription factor	Afu5g03920	*hapX*	Decreased conidiation and siderophore production	Decreased	[66]
Sterol regulatory element-binding protein	Afu2g01260	*srbA*	Inviable in anaerobic conditions	Decreased	[62]
Regulator of genes involved in gluconeogenesis	Afu2g12330	*acuM*	Decreased iron and carbon utilization	Decreased	[67]
Regulator of genes involved in gluconeogenesis	Afu2g05830	*acuK*	Decreased iron and carbon utilization	Decreased	[68]

**Table 3 jof-07-00518-t003:** *A. fumigatus* genes implicated in zinc metabolism and their implication for fungal phenotype and virulence.

Function	Systematic Name	Standard Name	Phenotype	Virulence	Ref.
Plasma membrane zinc transporter	Afu1g01550	*zrfA*	Decreased growth in Zn-limitation acid conditions	Unknown	[9]
Plasma membrane zinc transporter	Afu2g03860	*zrfB*	Decreased growth in Zn-limitation acid conditions	Unknown	[94]
Plasma membrane zinc transporter	Afu4g09560	*zrfC*	Decreased growth in Zn-limitation neutral/alkaline conditions	Decreased	[95]
Zinc C_2_H_2_ transcription factor	Afu1g10080	*zafA*	Decreased growth and germination in Zn-limitation	Decreased	[96]
pH-responsive C_2_H_2_ transcription factor	Afu3g11970	*pacC*	Decreased alkaline pH resistance	Decreased	[97]
Allergen	Afu4g09580	*aspf2*	Expressed in alkaline zinc-limiting conditions	Unknown	[95]
Zn2Cys6 binuclear transcription factor of Gliotoxin biosynthetic cluster	Afu6g09630	*gliZ*	Absence of gliotoxin production	Normal	[98]
Putative zinc importer	Afu6g00470	*zrfD*	Unknown	Unknown	
Putative zinc importer	Afu8g04010	*zrfE*	Unknown	Unknown	
Putative zinc importer	Afu2g08740	*zrfF*	Unknown	Unknown	
Putative zinc importer	Afu2g01460	*zrfG*	Unknown	Unknown	
Putative zinc importer	Afu2g12050	*zrfH*	Unknown	Unknown	
Putative zinc exporter	Afu1g12090	*zrgA*	Unknown	Unknown	
Putative zinc exporter	Afu6g14170	*mscA*	Unknown	Unknown	
Putative zinc exporter	Afu1g14440	*mtpA*	Unknown	Unknown	
Putative zinc exporter	Afu6g00440	*mtpB*	Unknown	Unknown	
Putative zinc exporter	Afu7g06570	*zrcA*	Unknown	Unknown	
Putative zinc exporter	Afu4g04150	*zrcB*	Unknown	Unknown	
Putative zinc exporter	Afu2g14570	*zrcC*	Unknown	Unknown	
Putative zinc exporter	Afu5g09830	*mmtA*	Unknown	Unknown	

## Data Availability

Data-sharing is not applicable to this article as no new data were created or analyzed in this study.

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
