# Peer review of "Nitrogen, Iron, and Zinc Acquisition: Key Nutrients to Aspergillus fumigatus Virulence"

_jof, 2021, doi:10.3390/jof7070518_

Round 1

Reviewer 1 Report

The authors present a detailed review of nitrogen, zinc and iron metabolism. In general the work is clear, however, I have provided multiple editorial suggestions (see attached file). Some additional specific comments:

  1. The Authors listed transcriptional regulator prtT as a gene implicated in nitrogen metabolism. However, prtT has also been associated with fungal iron metabolism. It has been reported to control expression of Aspergillus fumigatus iron uptake genes [doi:10.1371/journal.pone.0033604]. The authors need to address this point.
  2. Can Figure 2 with be a higher resolution image.
  3. In page 11 line 346-350, Pseudomonas aeruginosa siderophore pyoverdine has been mentioned as an anti-fungal agent for its role in iron sequestration. Pseudomonas produces another siderophore pyochelin, which has been reported to possess high affinity for zinc and the mode of antifungal effect includes iron and zinc deprivation to the fungus [doi:10.3390/jof5020048]. The authors need to include this while describing Aspergillus zinc metabolism.

Author Response

The authors present a detailed review of nitrogen, zinc and iron metabolism. In general the work is clear, however, I have provided multiple editorial suggestions (see attached file).

The editorial suggestions of attached file have been revised and the grammatical changes have been accepted. For more specific suggestion the following changes have been made:

  • Regarding comments on table titles and contents: The titles have been changed to: fumigatus genes implicated in x metabolism and their implication for fungal phenotype and virulence. (x = nitrogen, iron or zinc). In addition, blank boxes have been replaced for unknown.
  • Comment on Figure 1, consider labelling this as "proteases for nitrogen acquisition" otherwise seems disjointed from figure: We consider that is not necessary to add this term ‘proteases for nitrogen acquisition’ because PrtT is an transcription factor that upregulates protease secretion in general. The use of the derived peptides or amino acids by activities of these proteases are independent to PrtT action. Also, the derived forms may be used for carbon acquisition.

Comments of different phrases in text:

  • Line 176. We consider that the sentence is clear. The effect of areA deletion is important but not essential for fumigatus virulence. This gene is considered a virulence factor but in this sentence we only describe what happens with this mutant strain.
  • Line 191. Sorry for the mistake. It has been change for: “…neutropenic CD1 male mice”
  • Line 222. In this sentence we are referring to aerobiosis or anaerobiosis. To clarify we change to ‘type of atmosphere’ instead the suggestions of the reviewer.
  • Line 265. We have add the meaning of xenosiderophore as: “…siderophore used by an organism other than the one that produces it”
  • Line 280. This sentence is related to fungal development in any iron depleted environment, not only inside the human body so we consider that it is fine here.
  • Line 313. This has been modified in accordance with the suggestion of the reviewer as follows: “Moreover, although fumigatus is well adapted to cause human infection, it is not capable of uptaking iron from sources such as ferritin or transferrin [56] or in the case of hemoglobin only when its concentration is very high [82]”.
  • Line 328. We believe that indicating that there is no change in virulence with this gene is informative since if it is not indicated, it seems that it has not been studied.
  • Line 338. The sentence has been changed to: “Moreover, the immune response induced by the contact with WT or with siderophore non-producer strains in these cells was different.”
  • Line 343. This sentences was changed as: “the absence of HapX but not SreA attenuates virulence in the murine model of aspergillosis [70,71], underlining the critical role of adaptation to iron limitation in virulence.”
  • Line 346. The reviewer's suggestion is more specific and complete than the original sentence, thus we have substituted it.
  • Line 362. The sentence may be excessive long to a good compression so we have separated it in two different sentences: “In contrast, an excess of this metal causes toxicity by replacing other metals in protein-metalation sites. Although, zinc does not damage cells through ROS (as iron excess does) because this metal is redox-inactive.”
  • Line 368. We have considered that it is not necessary to join the two sentences, since joining sentences makes comprehension more difficult.
  • Line 392. These sentences are related with the growth of the strains in different environments not only in virulence.
  • Line 438. We refer to conditions during infection and sentence has been changed to clarify as follows: “… the infection process.”

Some additional specific comments:

  1. The Authors listed transcriptional regulator prtT as a gene implicated in nitrogen metabolism. However, prtT has also been associated with fungal iron metabolism. It has been reported to control expression of Aspergillus fumigatus iron uptake genes [doi:10.1371/journal.pone.0033604]. The authors need to address this point.
  • The reviewer is right. We have added the following sentence in line 463-464: “On the other hand, the protease regulator PrtT is able to upregulate iron uptake genes”. In addition, the corresponding paper has been included in as 106 reference.
  1. Can Figure 2 with be a higher resolution image.
  • The resolution of the Figure 2 has been changed from 300 to 600 ppp.
  1. In page 11 line 346-350, Pseudomonas aeruginosa siderophore pyoverdine has been mentioned as an anti-fungal agent for its role in iron sequestration. Pseudomonas produces another siderophore pyochelin, which has been reported to possess high affinity for zinc and the mode of antifungal effect includes iron and zinc deprivation to the fungus [doi:10.3390/jof5020048]. The authors need to include this while describing Aspergillus zinc metabolism.
  • Thank you for your comment. We have added the proposed reference [102] and the following sentences:

Lines 360: “In particular, this bacteria produces pyoverdine and pyochelin, iron chelators,…”

Lines 422-423: “Moreover, the bacterial competition interferes with availability of Zn since pioverdine and pyochelin siderophores are also Zn chelators.

Reviewer 2 Report

The manuscript by Perez-Cuesta et, al. reviews the molecular mechanisms for nitrogen and trace element acquisition by the fungal pathogen Aspergillus fumigatus.  The authors include relevant articles for a large number of gene products that have been investigated.  They also include factors identified by homology or bioinformatics to make the review the most up-to-date.  The topic is relevant to Aspergillus biology and particularly investigations of A. fumigatus virulence.  In this regard, the authors document nicely which factors have been tested for roles in virulence and which contribute to host infection by A. fumigatus.

The authors should be careful to describe which environment is being discussed (e.g., soil, host, in vitro media).  It is unclear why the authors include sections on iron and zinc, but not copper, which is also a key factor in nutritional immunity.  Other suggested corrections and additions are listed below.

line 16 "virulence-assayed" is not a good term.  Use "assayed for roles in virulence"

line 24 competitor with whom?

line 29 vast majority of inhaled conidia are cleared which doesn't equate to colonization or invasive aspergillosis

line 34  Catalases play a very minor role in pathogenesis.  The reference cited (Paris 2003) includes no mouse survival or fungal burden measurement of virulence, only reduced inflammation which may or may not affect virulence.

line 35  Missing key virulence factors such as GAG (Gravelat 2010, 2013) and superoxide dismutases (Lambou 2010) and adaptation to hypoxia (Grahl 2011, Willger 2012).  These have a more important role in virulence than the catalases.

line 80  glutamine synthetase doesn't "bind" the ammonia to glutamate

line 93  missing reference for pseudohyphal growth

line 210  nitrogen is already known to be an essential nutrient, the question is which nitrogen sources are relevant during host infection.

line 228  crucial for survival in the environment or in vivo - clarify statement

line 240  what is the source of electrons for iron reduction at the cell surface?

line 336 clarify conidia/germlings are the intracellular forms

line 368 the term "media" is incorrect as it depends on the in vitro media (whether Zn+ or Zn++ is provided).

line 402 give examples of host zinc-binding proteins

line 462 "basifying" is jargon, use "alkalinizing or neutralizing"

Author Response

The manuscript by Perez-Cuesta et, al. reviews the molecular mechanisms for nitrogen and trace element acquisition by the fungal pathogen Aspergillus fumigatus. The authors include relevant articles for a large number of gene products that have been investigated. They also include factors identified by homology or bioinformatics to make the review the most up-to-date. The topic is relevant to Aspergillus biology and particularly investigations of A. fumigatus virulence. In this regard, the authors document nicely which factors have been tested for roles in virulence and which contribute to host infection by A. fumigatus.

The authors should be careful to describe which environment is being discussed (e.g., soil, host, in vitro media). It is unclear why the authors include sections on iron and zinc, but not copper, which is also a key factor in nutritional immunity. Other suggested corrections and additions are listed below.

  • This comment is interesting but we think that cooper metabolism is already quite complex and large to include it.

line 16 "virulence-assayed" is not a good term. Use "assayed for roles in virulence"

  • This term has been changed

line 24 competitor with whom?

  • The sentence has been changed as: “… against other environmental microorganisms…”

line 29 vast majority of inhaled conidia are cleared which doesn't equate to colonization or invasive aspergillosis

  • The reviewer is right. To clarify, the sentence was change to: “… which reach the lungs, and may cause…”

line 34 Catalases play a very minor role in pathogenesis. The reference cited (Paris 2003) includes no mouse survival or fungal burden measurement of virulence, only reduced inflammation which may or may not affect virulence.

line 35 Missing key virulence factors such as GAG (Gravelat 2010, 2013) and superoxide dismutases (Lambou 2010) and adaptation to hypoxia (Grahl 2011, Willger 2012). These have a more important role in virulence than the catalases.

  • The reviewer is right. The sentences have been changed as follows: “Virulence factors previously described in the literature include genes involved in cell wall composition maintenance, such as melanin biosynthetic genes or Galactosaminogalactan; genes associated with combatting the immune system, such as superoxide dismutase

line 80 glutamine synthetase doesn't "bind" the ammonia to glutamate

  • This was changed as follows:Once inside the cytosol, the glutamine synthetase catalyzes the condensation of ammonia and glutamate to produce glutamine”.

line 93 missing reference for pseudohyphal growth

  • The references for pseudohyphal growth are 23 and 24 and we have included them above into the sentence but associated with each of the proteins involved, for greater precision.

line 210 nitrogen is already known to be an essential nutrient, the question is which nitrogen sources are relevant during host infection.

  • This is one of the keys that has not yet been studied in depth. We think that is not clear which nitrogen sources are relevant during infection process and the regulation processes may play a greater role that each these sources.

line 228 crucial for survival in the environment or in vivo - clarify statement

  • Iron homeostasis is essential in both because ROS generation and the inactivation of enzymes lead to fungal death. We believe that it is not necessary to indicate that is in all environments.

line 240 what is the source of electrons for iron reduction at the cell surface?

  • The metalloreductases are conserved among the species. However, the activity of these enzymes is not studied in The source of electrons may be cytosolic NADPH.

line 336 clarify conidia/germlings are the intracellular forms

  • The sentence has been changed as follows: “Siderophore biosynthesis is also related to growth of conidia inside host cells”

line 368 the term "media" is incorrect as it depends on the in vitro media (whether Zn+ or Zn++ is provided).

  • The reviewer is right. We have changed the term to environment.

line 402 give examples of host zinc-binding proteins

  • We have added two examples: p53 and Myt1

line 462 "basifying" is jargon, use "alkalinizing or neutralizing"

  • This term has been changed to alkalinizing